# Fragility Assessment of Traffic Embankments Exposed to Earthquake-Induced Liquefaction

**Aleš Oblak** [1,*], **Mirko Kosič** [2], **Antonio Viana Da Fonseca** [3] **and Janko Logar** [1]

1    Faculty of Civil and Geodetic Engineering, University of Ljubljana, Jamova cesta 2, 1000 Ljubljana, Slovenia; janko.logar@fgg.uni-lj.si
2    ZAG, Slovenian National Building and Civil Engineering Institute, Dimičeva ulica 12, 1000 Ljubljana, Slovenia; mirko.kosic@zag.si
3    CONSTRUCT-GEO, Faculty of Engineering, University of Porto (FEUP), Rua Dr. Roberto Frias, s/n, 4200-465 Porto, Portugal; viana@fe.up.pt
*    Correspondence: ales.oblak@fgg.uni-lj.si

**Abstract:** In this study, a vulnerability analysis of road and railway embankments to earthquake-induced liquefaction deformations was carried out. The result of the vulnerability analysis was a set of fragility curves that were obtained for several embankment and soil-profile geometries, as well as for the material properties of the liquefiable layer. The fragility curves were based on the numerical calculations obtained from FLAC 2D software in conjunction with the PM4Sand material model used for simulating the behavior of liquefaction-susceptible soils during dynamic shaking. The fragility analysis was performed using an incremental dynamic analysis approach considering a set of 30 ground motions and at least eight intensity levels. Permanent vertical displacement of the middle top point of the embankment was selected as the damage parameter, while the intensity measure was expressed in terms of peak ground acceleration at bedrock. Fragility curves were derived for three damage states, including minor, moderate and extensive damage, based on threshold values proposed in the literature. The influence of a single model variable was examined through comparison of the fragility curves.

**Keywords:** soil liquefaction; fragility curve; traffic embankment; FLAC; PM4Sand

## 1. Introduction

Natural disasters threaten human lives and possessions practically every day across the globe. According to Ritchie and Roser [1], most deaths (estimated at more than 800,000) from natural disasters in the last two decades were caused by earthquake events. In addition to the even greater socioeconomic losses due to flooding and extreme weather, losses caused by earthquakes in the last 20 years have exceeded more than US$ 530 billion. All of these losses resulted from either direct ground shaking or the various side effects of an earthquake event, such as a decrease in the shear strength of soil due to the occurrence of soil liquefaction in the ground, which can lead to the failure of infrastructure. Although it is difficult to distinguish the percentage of losses caused by liquefaction alone due to the diverse typology of buildings/engineering infrastructure and other features of the affected area, Youd et al. [2] mentioned that in case of the Costa Rica (1991) earthquake event, approximately 30% of the highway system of the affected area was damaged by liquefaction.

In the event of an earthquake, as well as in daily life, the flow capability of different lifelines is of vital importance. For example, the traffic network represents connections between living and work places, as well as between affected areas and assistant institutions in cases of natural disaster. The transportation infrastructure includes road and railway embankments (traffic embankments),

built for crossing diverse terrain or due to the needs of split-level junctions. When such engineering structures are located in seismically active areas and underlain by loose, sandy deposits, saturated with groundwater, severe damage can be caused by earthquake shaking and soil liquefaction.

Recent earthquakes around the world have indicated the vulnerability of transportation infrastructure to earthquake-induced liquefaction deformations [3–6]. Bird et al. [7] summarized the damage records of certain strong, historical earthquakes between 1989 and 2003 to traffic infrastructure and other structures due to soil liquefaction and ground shaking. In addition, Aryroudis et al. [8] reviewed the fragility assessment of the transport infrastructure subjected to multiple hazards.

In order to decrease the socioeconomic losses related to earthquake events and increase the resilience of society to these, the serviceability of traffic systems needs to be ensured as quickly as possible. With the intention of increasing our knowledge of the soil liquefaction phenomenon and its interaction with traffic embankments (particularly road and railway embankments), an extensive parametric study was conducted in this study. A vulnerability analysis of traffic embankments was performed, using various model geometries and material properties of the liquefiable layer. The result of the vulnerability analysis was a set of fragility curves, which can be used as an engineering tool to represent the probability of exceeding a certain damage state in relation to a selected intensity measure (IM) level. Since soil structure interaction is very complex in the case of soil liquefaction, different failure mechanisms of an embankment can occur, such as slipping of the slope surface, piping failure, crest settlements, lateral spreading, etc. [9–11]. Consequently, various engineering demand parameters (EDPs) have been used for the derivation of fragility curves. For example, Maruyama et al. [12] derived fragility curves based on damage datasets from the expressway embankments after a series of earthquakes in Japan. Lagaros et al. [13] used a factor of safety, while crest settlements are still widely used among other researchers as EDP [14–16], due to the simplicity of comparing these with field measurements. In addition to the different EDPs, fragility curves have been derived as a function of various IMs in these studies, such as with peak ground velocity [12], pseudostatic horizontal acceleration [13] and the more commonly used peak ground acceleration (PGA) [14–16]. In a slightly different context to earthquake-induced deformations of road embankments, McKenna and co-authors [17] studied the impact of groundwater level within the embankment and the size of scour at the toe of the embankment on the fragility curves, where the IM was defined as a function of embankment height and groundwater level, while crest settlements were taken as EDP.

Different approaches have been developed for the derivation of fragility curves, including empirical, analytical, expert judgement and hybrid methods [18]. Empirical fragility curves are based on extensive datasets from past earthquakes at locations with similar characteristics. The main problem with this approach is a lack of observed-damage surveys on particular objects under consideration for various seismic loading intensities. It is only appropriate for specific cases, and is not suitable for general situations. However, it does allow for soil–structure interaction to be observed in a more realistic way, compared to the other approaches, because local effects in the ground can be detected by advanced on-site measuring devices. In addition to the empirical method, rapid fragility-curve estimation can be achieved using expert judgement. This approach is convenient for an initial guesstimate of the situation only, since huge deviations among different experts are expected. On the other hand, use of the analytical method has been gaining popularity, with the improvement of computer and software capabilities, because it enables analysis of the same problem under different boundary and seismic loading conditions. The method allows a rigorous consideration of local seismological conditions. Nevertheless, all analytical (numerical) procedures are affected by modelling uncertainties, which means that all predictions are just an approximation of the reality. When one of the above methods cannot sufficiently cover the entire range of a fragility curve, the so-called hybrid method is used, in which several approaches are combined.

The following sections present a more detailed description of the methodologies used for the derivation of fragility curves, a description of the model geometries, soil parameters and input ground motions (GMs), and the final outcomes of the analyses. The main aim of the study is to show the

influence of different model variables on the response of an embankment built on liquefiable ground in terms of fragility curves. The study follows a known methodology for the derivation of fragility curves. Fragility of embankments considering earthquake-induced liquefaction has rarely been addressed by researchers. The contribution of Khalil et al. [16] studied the influence of isotropic and anisotropic permeability of the ground on the embankment response in terms of fragility curves. In our study, the influence of ground and embankment geometry (thickness of liquefiable layer, presence of crust layer and height and width of embankment) and of the relative density of the liquefiable layer on fragility curves was investigated.

## 2. Methodology for the Derivation of Fragility Curves

In general, vulnerability analysis is the process of systematically evaluating natural hazards and other threats that may potentially cause damage to an observed object. Among all available engineering tools, the fragility function is used to quantify the level of vulnerabilities to the object under consideration in terms of the probability of exceeding a certain damage state under a given IM.

The fragility function is commonly defined as a lognormal cumulative distribution function (Equation (1)), where *P(ds|IM = x)* is the probability that a GM with *IM = x* will exceed the selected damage state, $\Phi()$ is the standard normal cumulative distribution function, $\theta$ is the median of the fragility function (the IM level with a 50% probability of exceeding the damage state) and $\beta$ is the standard deviation of ln*IM* (sometimes referred to as the dispersion of IM) [19].

$$P(ds|IM = x) = \Phi\left(\frac{ln(x/\theta)}{\beta}\right) \tag{1}$$

Different procedures for performing numerical simulations to assemble the results needed for the derivation of fragility curves are available in the literature (e.g., [19]). One common approach is incremental dynamic analysis (IDA) [20], in which a set of GMs is scaled to find IM levels at which each GM causes a selected damage state. Alternatively, multiple stripe analysis can be used, in which the analysis is performed at a specified set of IM levels, each of which has a unique GM set [21,22].

In this study, the fragility analysis was performed by scaling a set of GMs at multiple IM levels. The employed procedure is conceptually consistent with IDA because the seismic response is obtained by the scaling of GMs. However, this approach does not involve an explicit estimation of the IM levels at which the GMs cause selected damage states. This was expected to yield a reduction in the number analyses compared to the conventional IDA.

### 2.1. Estimation of the Fragility Function Parameters

An example of the results of a seismic response assessment using the employed approach is presented in Figure 1a. As can be seen from the figure, the results can be used to estimate the conditional probability of exceeding a selected damage state at a given IM level, which is computed based on the ratio between the number of cases exceeding a selected damage state and the total number of GMs (Figure 1a). As suggested by Baker [19], the obtained results can be used to estimate the fragility parameters using the maximum likelihood method. Assuming the independence of the observations obtained for different GMs and a lognormal distribution of the fragility function (Equation (1)), the likelihood function is defined as the product of binomial probabilities at multiple IM levels, as follows:

$$Likelihood = \prod_{j=1}^{m} \binom{n_j}{z_j} \Phi\left(\frac{ln(x_j/\theta)}{\beta}\right)^{z_j} \left(1 - \Phi\left(\frac{ln(x_j/\theta)}{\beta}\right)\right)^{n_j-z_j} \tag{2}$$

where *m* denotes the number of IM levels, $z_j$ is the number of cases exceeding a certain damage state at the *m*th intensity level, $n_j$ is the number of used GMs, and $\prod$ is the product of all individual

likelihoods [19]. Estimates of the fragility function parameters are obtained by maximizing the likelihood function from Equation (2) or, equivalently, by maximizing the logarithm of the likelihood function as follows:

$$\{\hat{\theta}, \hat{\beta}\} = \underset{\theta, \beta}{\text{argmax}} \sum_{j=1}^{m} \left\{ ln\binom{n_j}{z_j} + z_j ln\, \Phi\left(\frac{ln(x_j/\theta)}{\beta}\right) + (n_j - z_j) ln\left(1 - \Phi\left(\frac{ln(x_j/\theta)}{\beta}\right)\right) \right\} \tag{3}$$

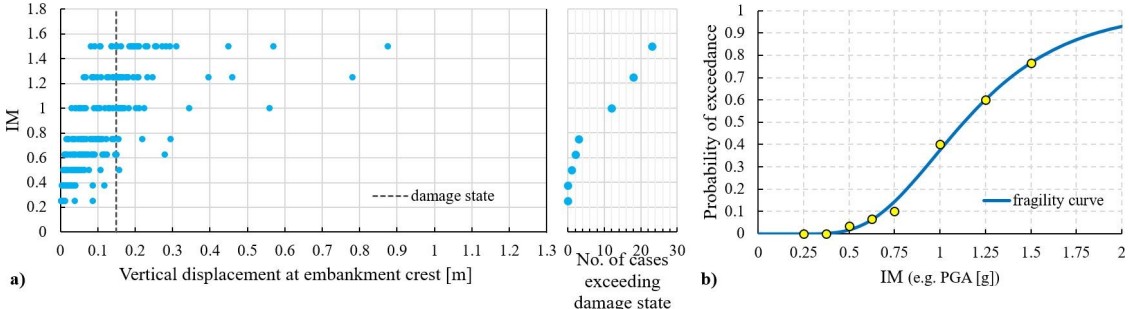

**Figure 1.** Example of the results of a seismic response assessment, with identification of the number of cases exceeding a selected damage state (**a**), and the corresponding fitted fragility curve (**b**).

An example of the fragility curve obtained using the above methodology is presented in Figure 1b.

It should be noted that the approach used to estimate fragility is somewhat approximate due to the use of the same set of GMs at different IM levels. However, the change in surface ground motion characteristics with increasing seismic intensity is partly taken into account by applying GMs at the base of the model (bedrock) and using non-linear material models for soil layers.

### 2.2. Definition of Damage States, EDP and IM

During dynamic excitation, additional stresses occur in the soil structure and embankments, leading to additional deformations. Even greater displacements are expected when the foundation ground is susceptible to liquefaction. Regarding various failure mechanisms of embankments (slope instability, crest settlements, lateral spreading, piping, etc.), different EDPs and their threshold values can be used in the analyses.

Apart from the characteristics of the seismic load, other factors can have a great impact on the extent of any deformations, including the height and shape of an embankment [23]. Moreover, Tang et al. [24] identified other parameters that control the seismic behavior of liquefiable soils, such as thickness of the liquefiable layer, stress history, depth of the sand layer and other site conditions, along with soil characteristics.

In this study, the permanent vertical displacement at embankment crest was used as the EDP in the derivation of the fragility curves. Threshold values for three damage states (ds1—minor, ds2—moderate and ds3—extensive) of the selected EDP were taken from the literature [14], and are equal to 0.05 m, 0.15 m and 0.40 m for road embankments and 0.03 m, 0.08 m and 0.20 m for railway embankments, respectively. In terms of serviceability, the damage state ds1 corresponds to open road/railway with speed reduction, while ds2 and ds3 to partially closed or fully closed traffic line during repair works. According to Argyroudis and Kaynia [14], the used limit values were obtained on the basis of expert opinion and were corroborated by the experiment driving test, in which the influence of road embankment settling in relation to driving speed was studied [25]. Additional damage state parameters with threshold values can be found in the literature (e.g., [12,13,18]).

PGA at bedrock was selected as the IMs for the derivation of the fragility curves in this research. This selection was based on the wide usage of PGA in practice, and its numerous correlations with other engineering parameters.

## 3. Model Description

### 3.1. Embankment and Soil Profile Geometry

The soil–structure interaction was studied on various embankment geometries, where the crest width and construction height were varied, while the slope inclination remained unchanged for a complete calculation set. The embankments were built on three- and, in the absence of a clayey crust layer, two-layered soil profiles with a horizontal ground surface. At the base, the stiff clayey layer is placed covered by a sandy layer that was susceptible to liquefaction. Four different ground profiles (Table 1) were constructed, including S1 with a 7 m thick liquefiable layer and a crust layer on top, S2 with a 7 m thick liquefiable layer without a crust layer, and S3 and S4 with a clayey crust and thickness of the sandy layer of 2 m and 4 m, respectively. A 24 m thick base layer was applied to all four soil profiles. A groundwater level ($z_w$ in Figure 2) was assigned 1 m below the free field surface in all four soil profiles.

**Table 1.** Ground profile variations.

| Soil ID | $H_C$—Thickness of Crust Layer [m] | $H_L$—Thickness of Liquefiable Layer [m] | $H_B$—Thickness of Base Layer [m] |
|---------|-----------------------------------|------------------------------------------|-----------------------------------|
| S1 | 1 | 7 | 24 |
| S2 | 0 | 7 | 24 |
| S3 | 1 | 2 | 24 |
| S4 | 1 | 4 | 24 |

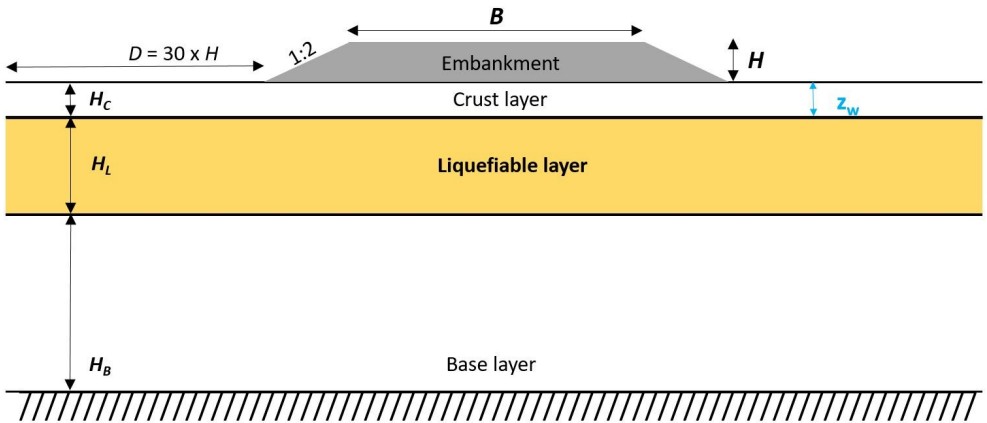

**Figure 2.** Variables.

Initially, three different crest widths were analyzed in relation to the 4 m high embankment underlain by soil profile S1; these were 6 m, 12 m and 24 m, representing a local road, a regional long-distance road and a motorway, respectively. In all further analyses, a 24 m wide embankment crest was adopted. The widest model embankment was selected in order to reduce the influence of the crest edges (slope instability) on the magnitude of settling at the midpoint of the embankment crest when studying the effect of embankment height. The effect of four different embankment heights (2 m, 4 m, 6 m and 8 m) was examined for all four soil profiles.

In total, 17 different model variations, regarding embankment shape and soil profile stratigraphy, were analyzed and are discriminated in Table 2. In addition, Figure 2 provides a graphical overview of the geometric variables of the model.

**Table 2.** Geometry variations.

| No. of Model Variations | 1 | 2 | 3 | 4 | 5 | 6 | 7 | 8 | 9 | 10 | 11 | 12 | 13 | 14 | 15 | 16 | 17 |
|---|---|---|---|---|---|---|---|---|---|---|---|---|---|---|---|---|---|
| Embankment height—H [m] | 4 | | 2 | | | | 4 | | | | 6 | | | | 8 | | |
| Crest width—B [m] | 6 | 12 | 24 | | | | | | | | | | | | | | |
| Soil profile ID | S1 | | S1 | S2 | S3 | S4 | S1 | S2 | S3 | S4 | S1 | S2 | S3 | S4 | S1 | S3 | S4 |

## 3.2. Material Properties

The selection of the material parameters (Table 3) used in the numerical simulations was based on the results of ground investigations from the Lisbon area within the Liquefact project [26]. The notations for the material properties are explained below the table.

**Table 3.** Material properties.

| Layer | $\rho_{dry}$ [kg/m³] | $k$ [m/s] | Mohr–Coulomb | | | | | PM4Sand | | |
|---|---|---|---|---|---|---|---|---|---|---|
| | | | $K$ [MPa] | $G$ [MPa] | $c_u$ [kPa] | $\varphi'$ [°] | $c'$ [kPa] | $D_r$ [/] | $G_0$ [/] | $h_{po}$ [/] |
| Crust layer | 1784 | $8 \cdot 10^{-8}$ | 64 | 30 | 80 | | | | | |
| Liquefiable layer—medium dense | 1486 | $1.6 \cdot 10^{-5}$ | / | / | / | / | / | 0.60 | 760 | 0.55 |
| Liquefiable layer—loose | 1486 | $1.6 \cdot 10^{-5}$ | / | / | / | / | / | 0.35 | 476 | 0.5 |
| Base layer | 1436 | $1 \cdot 10^{-9}$ | 227 | 105 | 150 | / | / | / | / | / |
| Embankment | 1800 | $1.18 \cdot 10^{-5}$ | 83.3 | 38.5 | / | 35 | 5 | / | / | / |

Notations: $\rho_{dry}$—dry density, $k$—soil permeability, $K$—bulk modulus, $G$—shear modulus, $c_u$—undrained shear strength, $\varphi'$—friction angle, $c'$—cohesion, $D_r$—relative density, $G_0$—shear modulus coefficient, $h_{po}$—contraction rate parameter.

The upper crust layer and base layer represent clay, with an undrained shear strength equal to 80 kPa and 150 kPa, respectively. Both clayey layers were considered to be resistant to pore pressure build-up during seismic excitation. Analyses were performed for two sets of material properties, regarding the density of the sandy layer. A relative density of $D_r = 0.6$ was used to represent a medium density state, while $D_r = 0.35$ was assigned as a loose state. Depending on the corrected SPT blow count $(N_1)_{60}$ and an adopted value originating from the development of Boulanger and Ziotopoulou's [27] liquefaction triggering correlations ($C_d = 46$), the relative density, $D_r$, and shear modulus coefficient, $G_0$, can be estimated using Equations (4) and (5), as proposed by Boulanger and Ziotopoulou [27]:

$$D_r = \sqrt{\frac{(N_1)_{60}}{C_d}} \tag{4}$$

$$G_0 = 167\sqrt{(N_1)_{60} + 2.5} \tag{5}$$

The PM4Sand material model [27] was assigned to the sandy layer, which was susceptible to liquefaction. The Mohr–Coulomb (MC) material model was used for the embankment and the two clayey layers.

## 3.3. 2D Numerical Model

The numerical simulations in this study were performed with FLAC 2D software package in conjunction with the advanced material model PM4Sand, which was developed for simulating the progress of earthquake liquefaction-induced deformations. The model can be calibrated with elementary or more advanced laboratory and in-situ tests with a reasonable amount of engineering effort and has been successfully validated in the literature [27–29]. The advantage of the material model was that it supports the simultaneous calculation of pore water pressure build-up due to seismic excitation and the corresponding soil densification and its dissipation [27].

The FLAC 2D software package was used for the numerical simulations following a successful validation of the modelling technique being demonstrated in the case of liquefaction-induced deformations of the Naruse River levee in Japan [30]. When comparing the results of the numerical simulation to field measurements from on and under the Naruse embankment, the magnitude of the final displacements and the failure mechanism itself, as well as the increase in excess pore water pressure through the time of dynamic loading in a liquefiable layer, were satisfactorily captured.

Initially, the ground stratigraphy, with a groundwater level and hydrostatic pore water pressure distribution, soil parameters and boundary conditions for static calculation, were applied successively in the model. Vertical displacements at the sides were allowed, while movements at the base were restricted in both the vertical and horizontal directions for this modelling phase. In addition, water drainage was assumed only through the model surface. In following phases, the embankment was constructed on the free field ground, the material model was modified in the liquefiable layer from the MC to the advanced PM4Sand (the MC material model remained assigned to the other, non-liquefiable layers), and the boundary conditions were transformed, consistent with the dynamic analysis requirements. A compliant base was adopted at the bottom of the model, and free field boundary conditions were adopted at the sides. Since the soil damping and free field boundary conditions themselves could not ensure complete absorption of the reflected seismic waves at the sides, the model width was selected as a function of embankment height (see Figure 2). In this way, the influence of the lateral edges on embankment behavior was negligible. Moreover, a small fraction (0.5%) of Rayleigh damping, operating around a frequency equal to 5 Hz, was assigned to the model. Since we were primarily interested in deformations in the embankment caused by seismic loading, the displacements calculated in the previous phases were set to zero before the shear stress history (which was calculated from the acceleration time history) was applied at the bottom of the model. Accurate wave transmission through the soil profile was achieved using a selection of grid-size distributions, by refining the $1 \times 1$ m mesh size at the center of the model, under the embankment, while element width was gradually increased towards the model edges.

### 3.4. Ground Motions

For the purpose of the fragility analysis, a set of 30 GMs was selected from the strong-motion database that combines recordings from the Next Generation Attenuation (NGA) [31] and from the Reference database for seismic ground-motion prediction in Europe (RESORCE) database [32]. The selection of GMs was performed according to the procedure proposed by Jayaram et al. [33], considering the following constraints regarding the magnitude ($M$), source-to-site distance ($R$), and shear-wave velocity in the upper 30 m of soil ($v_{s,30}$): $5.5 < M < 7.5$; $5$ km $< R < 50$ km; and $v_{s,30} > 500$ m/s. The maximum scale factor in the ground-motion selection was limited to 2.0. The GMs were selected in such a way that the mean of the horizontal acceleration spectra matched the elastic spectrum according to Eurocode 8 (EC8: [34]) for soil type A by being conditioned to the peak ground acceleration of 0.25g. Note that the EC8's criterion [34] for the $v_{s,30}$ of soil type A ($v_{s,30} > 800$ m/s) had to be relaxed due to the limited number of GMs recorded from rock outcrop or rock-equivalent stiff soil. Figure 3 shows a comparison of the EC8 spectrum for soil class A and the mean spectrum obtained for the selected set of 30 GMs. The duration of the selected GMs varied between 7.4 s and 61.0 s (average duration of 17.4 s).

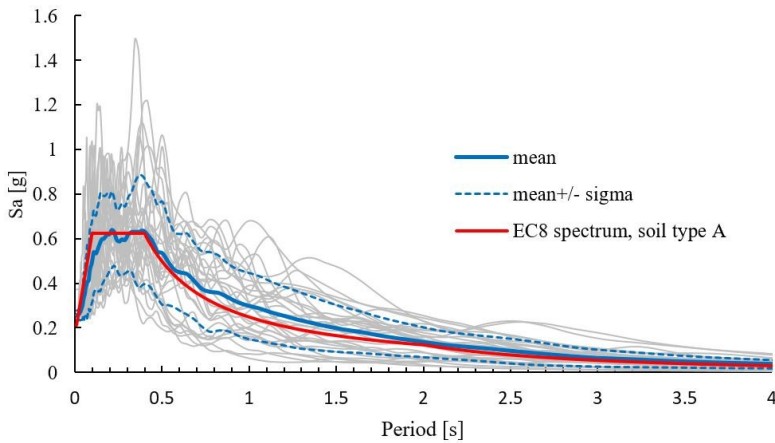

**Figure 3.** Response spectra (5% damping) of the selected set of GMs used in the fragility analysis.

The decision on the number of GMs was based on a literature review. The typical number of GMs for the estimation of the distribution of structural response ranged between 14 and 200, but most of the studies considered 20 to 40 GMs [35]. Recently, Scozzese et al. [36] recommended using 20 GMs to obtain sufficient accuracy of the results in the case of multiple stripe analysis.

In order to cover the entire range of fragility curves, the GMs were scaled to eight intensity levels in terms of maximum peak ground acceleration at bedrock (0.25 g, 0.375 g, 0.5 g, 0.625 g, 0.75 g, 1.0 g, 1.25 g and 1.5 g). Additional intensity levels (0.1 g, 0.15 g, 0.2 g and 0.3 g) were used in the analyses with loose sand, where the threshold values of crest settlements were exceeded in the majority of performed cases, even at low intensities.

## 4. Results and Discussion

Due to the large amount of data that resulted from the performed analyses, the time histories of the displacements, accelerations and pore pressure ratios were recorded only at selected points. The locations of the recording points are presented in Figure 4. The embankment displacements were recorded at the midpoint and at the edges of the embankment contour. The horizontal accelerations were recorded at the midpoint of the embankment at ground level and at the bottom of the model for the acceleration time histories, whereas the pore pressure ratios were recorded at three points at the center of the liquefiable layer (at the midpoint and below the embankment toes).

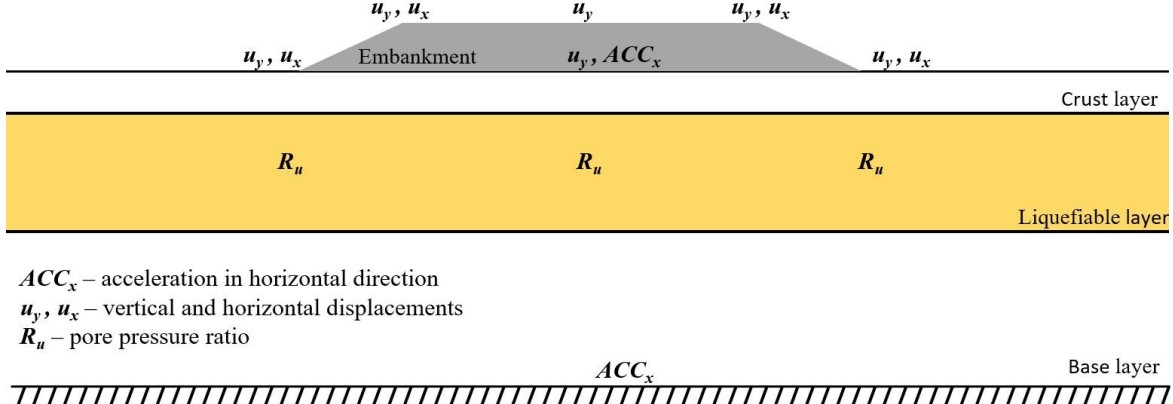

**Figure 4.** Locations of the selected recording points.

### 4.1. Displacements, Accelerations and Pore Pressure Ratio Time Histories

A representative deformed shape of the embankment, along with the location of the recording point of the vertical displacement used for the definition of the embankment damage state, is presented in Figure 5, for the case of a 6 m high embankment, underlain by soil profile S1 and subjected to a PGA at bedrock of 0.6 g. In the literature [37,38], a similar failure mechanism was observed as was seen in the embankments' earthfill material considered in this study. In general, ground liquefaction causes lateral movements that result in rotational slope failure and crest settling. The amount and magnitude of such displacements depend on many factors, including GM intensity, embankment geometry and soil parameters. The majority of the deformations of the embankment occurred during a strong part of the GM, when the shaking had the greatest impact and liquefaction was fully developed. Later, the displacement increments gradually decreased and the displacements settled at their end values.

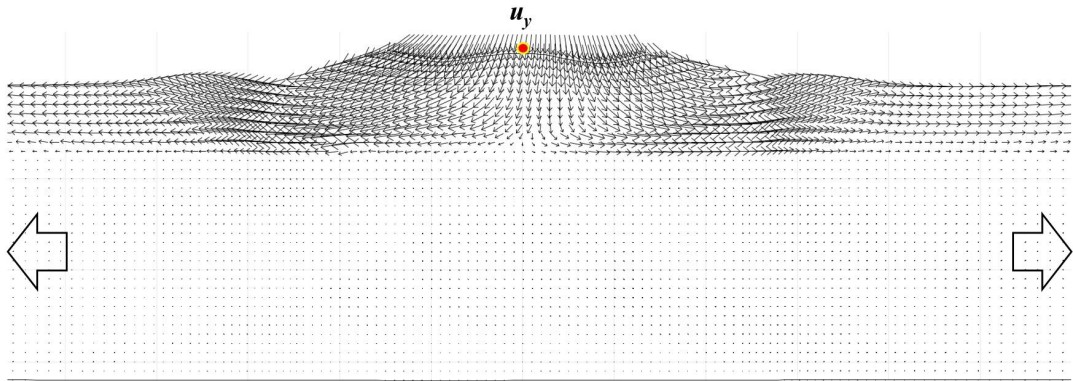

**Figure 5.** Example of a deformed embankment shape ($u_y$—vertical displacement at the crest midpoint used for derivation of fragility curves).

For the same example presented in Figure 5, the displacement time history at the crest's midpoint, the pore pressure ratio ($R_u$) distribution in the middle of the liquefiable layer under the embankment's longitudinal axis and the acceleration time history under the embankment and at the base of the model are shown in Figure 6.

From Figure 6b, it can be seen that the pore pressure ratio rose smoothly until it rapidly (in a second or two) reached unity during the strong part of the ground shaking. Afterwards, it gradually decreased with the weakening of the dynamic load. Combining these observations with the embankment deformations, it can be seen that during the same part of the ground motion, the majority of the displacement at the crest midpoint occurred (Figure 6a). When examining the results of all the performed analyses, it was apparent that the dissipation of excess pore water pressure was highly dependent on the soil parameters (e.g., permeability) and other drainage conditions. For example, when comparing the results obtained for soil profiles S1 and S2, both with the same thickness of liquefiable layer, it was noted that a faster dissipation and thus lower $R_u$ at the end of shaking were obtained for soil profile S2 due to the absence of the clayey crust layer with its low permeability.

Furthermore, a possible de-amplification effect on the acceleration time histories can be observed under the embankment (Figure 6c), since sandy soils that turn into a liquid mixture during liquefaction are incapable of transmitting shear stresses to the upper layers.

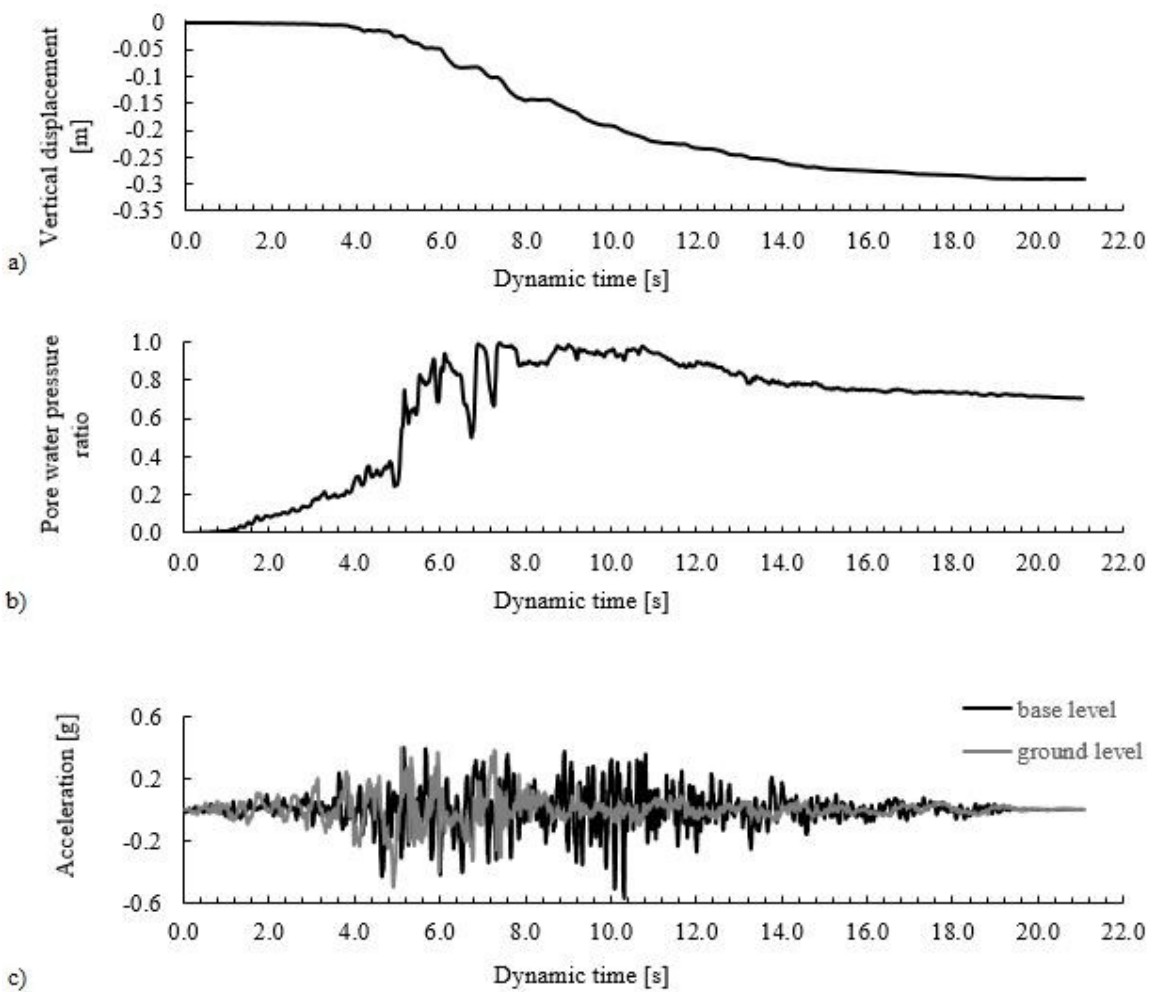

**Figure 6.** Vertical crest displacement (**a**), pore pressure ratio (**b**) and acceleration time (**c**) histories.

## 4.2. Fragility Curves

The complete set of results for all studied examples are given in Tables 4–9 in terms of median value of fragility function ($\theta$) and standard deviation of ln$IM$ ($\beta$). These two fragility-curve parameters completely define each fragility curve (see Equation (1)). It must be emphasized, however, that these results are only valid within the confines of the assumptions made in this study (e.g., material properties, geometry of ground layers and embankment) and cannot be readily extrapolated for other conditions. The main scope of the study was to show the influence in individual model variables on fragility curves.

**Table 4.** Fragility curve (FC) parameters for different crest widths (medium dense soil, peak ground acceleration (PGA), road criteria).

| Medium Dense Case ($D_r$ = 0.6) | Soil ID | Fragility Curve Parameters | Crest Width (Embankment Height *H* = 4 m) | | | | | | | | |
|---|---|---|---|---|---|---|---|---|---|---|---|
| | | | 6 m | | | 12 m | | | 24 m | | |
| | | | ds1 | ds2 | ds3 | ds1 | ds2 | ds3 | ds1 | ds2 | ds3 |
| | S1 | $\theta$ | 0.35 | 0.56 | 1.13 | 0.38 | 0.58 | 1.21 | 0.42 | 0.71 | 1.54 |
| | | $\beta$ | 0.31 | 0.40 | 0.43 | 0.33 | 0.44 | 0.47 | 0.42 | 0.33 | 0.51 |

**Table 5.** FC parameters for different embankment heights (medium dense soil, PGA, road criteria).

| | Soil ID | Fragility Curve Parameters | Embankment Height (Crest Width *B* = 24 m) | | | | | | | | | | | |
| --- | --- | --- | --- | --- | --- | --- | --- | --- | --- | --- | --- | --- | --- | --- |
| | | | 2 m | | | 4 m | | | 6 m | | | 8 m | | |
| | | | ds1 | ds2 | ds3 | ds1 | ds2 | ds3 | ds1 | ds2 | ds3 | ds1 | ds2 | ds3 |
| **Medium Dense Case (*D$_r$* = 0.6)** | S1 | $\theta$ | 0.49 | 0.89 | / | 0.42 | 0.71 | 1.54 | 0.41 | 0.68 | 1.20 | 0.41 | 0.59 | 1.05 |
| | | $\beta$ | 0.37 | 0.52 | / | 0.42 | 0.33 | 0.51 | 0.42 | 0.40 | 0.35 | 0.38 | 0.39 | 0.36 |
| | S2 | $\theta$ | 0.42 | 0.77 | / | 0.39 | 0.62 | 1.37 | 0.40 | 0.59 | 1.08 | | | |
| | | $\beta$ | 0.37 | 0.37 | / | 0.34 | 0.39 | 0.50 | 0.37 | 0.41 | 0.41 | | | |
| | S3 | $\theta$ | / | / | / | 0.83 | <span style="color:red">1.80</span> | / | 0.49 | 0.72 | 1.50 | 0.52 | 0.76 | 1.20 |
| | | $\beta$ | / | / | / | 0.44 | <span style="color:red">0.45</span> | / | 0.40 | 0.39 | 0.54 | 0.33 | 0.37 | 0.41 |
| | S4 | $\theta$ | 1.03 | / | / | 0.60 | 1.13 | / | 0.45 | 0.80 | 1.48 | 0.45 | 0.62 | 1.01 |
| | | $\beta$ | 0.49 | / | / | 0.37 | 0.39 | / | 0.36 | 0.40 | 0.40 | 0.42 | 0.38 | 0.38 |

**Table 6.** FC parameters for different embankment heights (loose soil, PGA, road criteria).

| | Soil ID | Fragility Curve Parameters | Embankment Height (Crest Width *B* = 24 m) | | | | | | | | | | | |
| --- | --- | --- | --- | --- | --- | --- | --- | --- | --- | --- | --- | --- | --- | --- |
| | | | 2 m | | | 4 m | | | 6 m | | | 8 m | | |
| | | | ds1 | ds2 | ds3 | ds1 | ds2 | ds3 | ds1 | ds2 | ds3 | ds1 | ds2 | ds3 |
| **Loose Case (*D$_r$* = 0.35)** | S1 | $\theta$ | 0.21 | 0.37 | 1.03 | 0.18 | 0.31 | 0.68 | 0.18 | 0.30 | 0.59 | 0.18 | 0.24 | 0.50 |
| | | $\beta$ | 0.40 | 0.38 | 0.43 | 0.40 | 0.39 | 0.40 | 0.42 | 0.35 | 0.42 | 0.40 | 0.41 | 0.45 |
| | S2 | $\theta$ | 0.17 | 0.31 | 0.86 | 0.17 | 0.26 | 0.56 | 0.17 | 0.24 | 0.49 | | | |
| | | $\beta$ | 0.37 | 0.41 | 0.41 | 0.38 | 0.40 | 0.42 | 0.39 | 0.43 | 0.46 | | | |
| | S3 | $\theta$ | <span style="color:red">0.84</span> | / | / | 0.23 | 0.54 | / | 0.20 | 0.38 | 0.87 | 0.18 | 0.29 | <span style="color:red">0.81</span> |
| | | $\beta$ | <span style="color:red">1.03</span> | / | / | 0.35 | 0.53 | / | 0.25 | 0.48 | 0.46 | 0.37 | 0.39 | <span style="color:red">0.69</span> |
| | S4 | $\theta$ | 0.35 | 1.15 | / | 0.22 | 0.50 | 1.26 | 0.19 | 0.40 | 0.80 | 0.19 | 0.27 | 0.55 |
| | | $\beta$ | 0.34 | 0.61 | / | 0.35 | 0.34 | 0.42 | 0.42 | 0.39 | 0.38 | 0.39 | 0.37 | 0.43 |

**Table 7.** FC parameters for different crest widths (medium dense soil, PGA, railway criteria).

| | Soil ID | Fragility Curve Parameters | Crest Width (Embankment Height *H* = 4 m) | | | | | | | | |
| --- | --- | --- | --- | --- | --- | --- | --- | --- | --- | --- | --- |
| | | | 6 m | | | 12 m | | | 24 m | | |
| | | | ds1 | ds2 | ds3 | ds1 | ds2 | ds3 | ds1 | ds2 | ds3 |
| **Medium Dense Case (*D$_r$* = 0.6)** | S1 | $\theta$ | 0.31 | 0.42 | 0.70 | 0.37 | 0.51 | 0.85 | 0.38 | 0.51 | 0.85 |
| | | $\beta$ | 0.30 | 0.42 | 0.45 | 0.33 | 0.37 | 0.35 | 0.35 | 0.40 | 0.36 |

**Table 8.** FC parameters for different embankment heights (medium dense soil, PGA, railway criteria).

| | Soil ID | Fragility Curve Parameters | Embankment Height (Crest Width *B* = 24 m) | | | | | | | | | | | |
| --- | --- | --- | --- | --- | --- | --- | --- | --- | --- | --- | --- | --- | --- | --- |
| | | | 2 m | | | 4 m | | | 6 m | | | 8 m | | |
| | | | ds1 | ds2 | ds3 | ds1 | ds2 | ds3 | ds1 | ds2 | ds3 | ds1 | ds2 | ds3 |
| **Medium Dense Case (*D$_r$* = 0.6)** | S1 | $\theta$ | 0.44 | 0.60 | 1.23 | 0.38 | 0.51 | 0.85 | 0.36 | 0.49 | 0.82 | 0.35 | 0.46 | 0.71 |
| | | $\beta$ | 0.40 | 0.33 | 0.59 | 0.35 | 0.40 | 0.36 | 0.32 | 0.41 | 0.37 | 0.27 | 0.40 | 0.38 |
| | S2 | $\theta$ | 0.37 | 0.52 | 1.09 | 0.34 | 0.43 | 0.77 | 0.34 | 0.45 | 0.69 | | | |
| | | $\beta$ | 0.31 | 0.38 | 0.47 | 0.28 | 0.36 | 0.38 | 0.30 | 0.41 | 0.39 | | | |
| | S3 | $\theta$ | / | / | / | 0.65 | 1.15 | / | 0.42 | 0.57 | 0.86 | 0.42 | 0.60 | 0.85 |
| | | $\beta$ | / | / | / | 0.37 | 0.45 | / | 0.37 | 0.33 | 0.45 | 0.43 | 0.33 | 0.39 |
| | S4 | $\theta$ | 0.67 | 1.53 | / | 0.46 | 0.74 | <span style="color:red">1.56</span> | 0.42 | 0.61 | 0.96 | 0.43 | 0.49 | 0.73 |
| | | $\beta$ | 0.38 | 0.56 | / | 0.33 | 0.40 | <span style="color:red">0.47</span> | 0.38 | 0.35 | 0.37 | 0.39 | 0.39 | 0.42 |

**Table 9.** FC parameters for different embankment heights (loose soil, PGA, railway criteria).

| | Soil ID | Fragility Curve Parameters | Embankment Height (Crest Width *B* = 24 m) | | | | | | | | | | | |
| --- | --- | --- | --- | --- | --- | --- | --- | --- | --- | --- | --- | --- | --- | --- |
| | | | 2 m | | | 4 m | | | 6 m | | | 8 m | | |
| | | | ds1 | ds2 | ds3 | ds1 | ds2 | ds3 | ds1 | ds2 | ds3 | ds1 | ds2 | ds3 |
| Loose Case (*D_r* = 0.35) | S1 | $\theta$ | 0.19 | 0.24 | 0.47 | 0.16 | 0.21 | 0.38 | 0.16 | 0.21 | 0.37 | 0.16 | 0.21 | 0.30 |
| | | $\beta$ | 0.34 | 0.40 | 0.45 | 0.37 | 0.43 | 0.37 | 0.36 | 0.40 | 0.36 | 0.38 | 0.43 | 0.39 |
| | S2 | $\theta$ | 0.16 | 0.21 | 0.40 | 0.15 | 0.18 | 0.32 | 0.15 | 0.19 | 0.29 | | | |
| | | $\beta$ | 0.34 | 0.38 | 0.41 | 0.34 | 0.44 | 0.41 | 0.34 | 0.39 | 0.41 | | | |
| | S3 | $\theta$ | 0.30 | / | / | 0.18 | 0.30 | 0.77 | 0.19 | 0.23 | 0.45 | 0.18 | 0.22 | 0.36 |
| | | $\beta$ | 0.96 | / | / | 0.33 | 0.38 | 0.46 | 0.26 | 0.39 | 0.47 | 0.35 | 0.38 | 0.46 |
| | S4 | $\theta$ | 0.27 | 0.48 | / | 0.18 | 0.32 | 0.62 | 0.17 | 0.23 | 0.47 | 0.18 | 0.21 | 0.34 |
| | | $\beta$ | 0.32 | 0.41 | / | 0.38 | 0.29 | 0.36 | 0.39 | 0.34 | 0.41 | 0.38 | 0.37 | 0.40 |

Some cells in Tables 4–9 are empty (/) or have values highlighted in red, indicating less reliable results, because the adopted damage state criteria were either too strict or too flexible with respect to the range of intensity levels considered in this study. The number of exceedances was either very low over the entire intensity range or, on the other hand, the limit states were exceeded at very low intensities. In both cases, the fragility curve could not be properly derived.

Tables 4–9 summarize fragility curve parameters for both damage state criteria (roads and railways) and all performed analyses, where peak ground acceleration at bedrock was used as IM for the derivation of fragility curves.

Some of the obtained fragility curves were verified on the case history of Cark Canal site, located in Adapazari, Turkey [30]. In general, the site was strongly affected by the 1999 Kocaeli Earthquake, but at the location of the Cark Canal river embankments no sign of complete failure or heavy destruction was observed. According to Bay and Cox [39] and Youd et al. [40], no severe ground deformations (displacements not exceeding 10 cm) occurred at the site for a 4–5 m high river embankment lying on the 1–1.5 m thick liquefiable layer covered by a clayey crust. The same conclusions are reached with the proposed fragility curves (similar geometry to our model variation 9, medium dense soil, road criteria), where it was shown that the probability of damage due to liquefaction for a bedrock acceleration of 0.4 g corresponds to an exceedance probability of approximately 5–8% for minor damage state (ds1) (probability of experiencing less sever or equal damage to ds1, i.e., ground deformations below or equal to 5 cm, is thus around 95%), which is in agreement with the field evidence.

Typical examples of a fragility curve response to variations in the selected model variables are presented in Figures 7–10. The trends observed in changes in the fragility curves with variation in the model parameters (e.g., embankment height, thickness of liquefiable layer) are similar for the damage state criteria of roads and railways. Therefore, in some figures, the damage state criteria for roads were used and in the others, the damage state criteria for railways were used.

Fragility curves were derived for three different damage states representing minor (ds1), moderate (ds2) and significant (ds3) damage to the road/railway infrastructure, due to settlements at the embankment crest. Light green, blue and red colors were selected to differentiate between these three damage states, while for the other model variations (width and height of the embankment, thickness and relative density of the liquefiable layer, presence of crust layer), various line types were used in the figures.

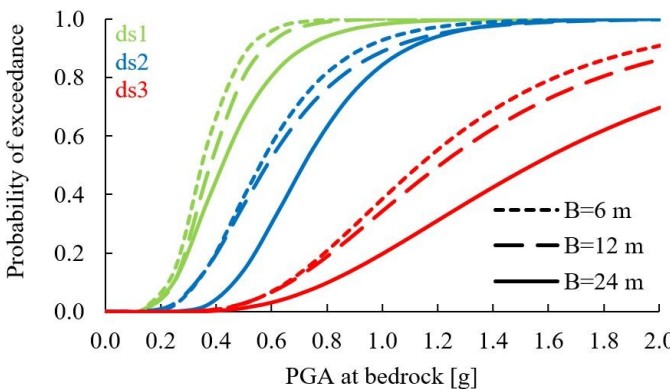

**Figure 7.** Influence of embankment width on fragility curves (damage state criteria for road embankments).

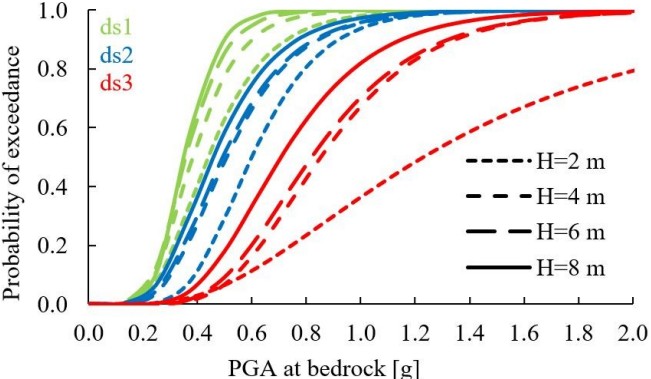

**Figure 8.** Influence of embankment height on fragility curves (damage state criteria for railway embankments).

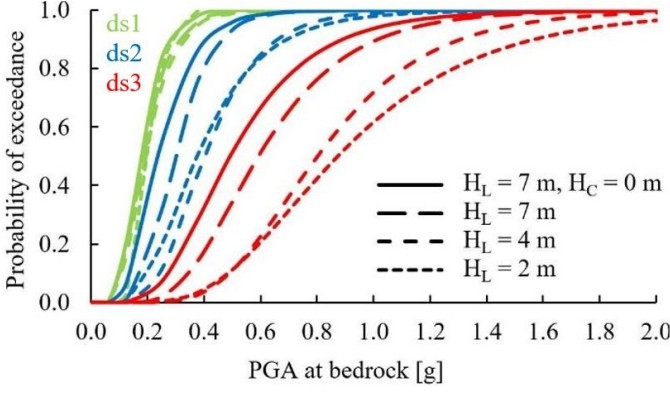

**Figure 9.** Influence of thickness of liquefiable layer on fragility curves (damage state criteria for road embankments).

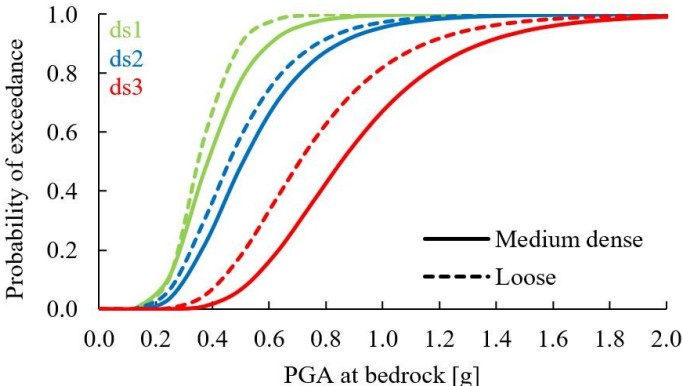

**Figure 10.** Influence of density of liquefiable layer on fragility curves (damage state criteria for railway embankments).

Initially, the influence of embankment width was studied for the case with a 4 m high embankment, situated on soil profile S1 (7 m thick liquefiable layer beneath a 1 m thick clayey crust layer) for a medium dense state of the sandy soil (Figure 7). From the figure, it is clear that the displacement of the central point of the embankment crest for this particular case decreased with an increase in embankment width, since the impact of less stable slopes was smaller. Consequently, the curves for the wider embankments move to the right, indicating lower vulnerability. All further analyses were performed using 24 m wide embankment crests.

The effect of embankment height on settlements at the top midpoint is expressed by the fragility curves in Figure 8. These curves were derived for 2 m, 4 m, 6 m and 8 m high embankments, situated above soil profile S1 and containing a medium-density liquefiable layer. The probability of exceeding the examined limit states increased with increasing height of the embankment.

There was a similar impact on the fragility curves with an increase in the thickness of the liquefiable layer, for the examined range in the study (2, 4 and 7 m). The greater the thickness of the liquefiable layer, the larger the vertical displacement at the crest. Consequently, there was a higher-numbered limit-state exceedance at each intensity, causing a shift of the fragility curves to the left. Even larger displacements were obtained in the absence of the crust layer ($H_C$ = 0 m). The effect of the thickness of the sandy layer is presented in Figure 9 for the case of a 6 m embankment and loose sandy material. The curves were derived on the basis of damage state criteria for road embankments using the PGA at bedrock as the IM. Due to the relatively high embankment and the loose density state of the liquefiable layer, the minimum threshold value was exceeded rapidly in all the soil profiles, which also caused a significant overlap of the curves for damage state ds1 (Figure 9).

Figure 10 shows the fragility curves for medium-density and loose state liquefiable layers. A comparison was made with a model with a 4 m thick sandy layer beneath a 1 m thick crust layer (soil profile S4) and a 6 m high embankment on top. As expected, with the embankments situated above saturated, loose sand deposits are highly vulnerable because liquefaction can develop to its full potential in loose soil, even at lower earthquake intensities.

## 4.3. Interdependencies between Possible EDPs

Although the permanent vertical displacement of an embankment crest has been proposed in the literature ([14]) as an EDP for the derivation of fragility curves, many other damage indicators could more accurately characterize the state of damage for an embankment after a seismic event. For example, since lateral spreading represents the main failure mechanism, traffic infrastructure could be predominantly damaged by lateral movements or by a difference in horizontal displacements either side of an embankment. Despite this well-known observation, no widely acceptable and experimentally supported threshold values related to horizontal displacement were found in the literature. In order to validate the use of vertical displacement at the top midpoint of an embankment ($u_{y\_mid}$) as an EDP,

we studied its correlation with three other possible damage state indicators ($u_{y\_edge}$, $|\Delta u_y|$ and $\Delta u_x$), as defined by following equations:

$$u_{y\_edge} = -\max\left(\left|u_{y\_right}\right|, \left|u_{y\_left}\right|\right)) \tag{6}$$

$$\left|\Delta u_y\right| = \max\left(\left|u_{y\_mid} - u_{y\_left}\right|, \left|u_{y\_mid} - u_{y\_right}\right|\right) \tag{7}$$

$$\Delta u_x = u_{x\_right} - u_{x\_left} \tag{8}$$

where $u_{y\_right}$, $u_{y\_mid}$ and $u_{y\_left}$ are the maximum vertical displacements at the crest right edge, midpoint and left edge, respectively, and $u_{x\_right}$ and $u_{x\_left}$ are the maximum horizontal displacements at the crest right and left edges, respectively.

The correlations between permanent embankment midpoint settlements ($u_{y\_mid}$) and the three alternative damage state parameters are presented in Figures 11 and 12. Nearly 8000 data points from the final calculated state of the numerical simulations are contained in these three graphs, including the results for four different embankment heights, four soil profiles, two density states of the liquefiable layer and 30 GMs with various intensities.

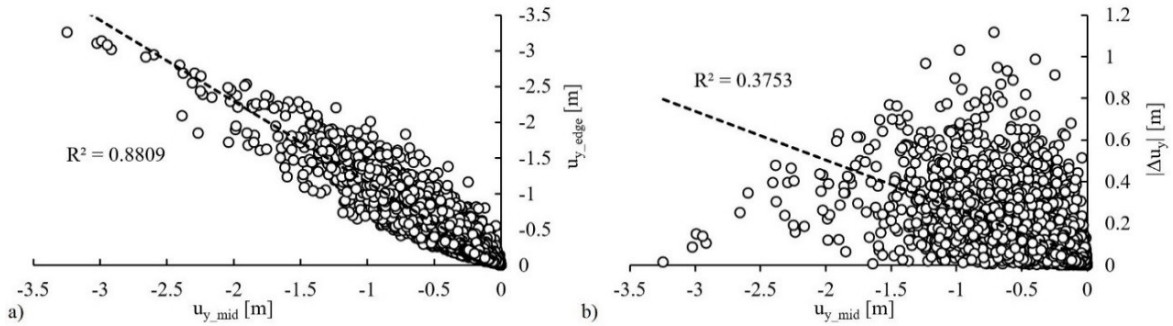

**Figure 11.** Correlation of $u_{y\_mid}$ to (**a**) vertical displacements at crest edges and (**b**) absolute differences between edge and middle crest points.

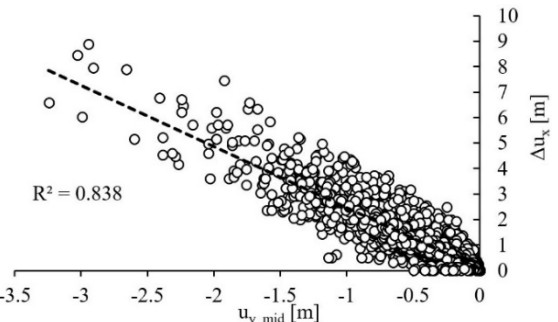

**Figure 12.** Correlation of $u_{y\_mid}$ to horizontal differential displacements.

According to the observed scatter and coefficient of determination, $R^2$, in Figures 11 and 12, it can be seen that both $u_{y\_mid}$ and $u_{y\_edge}$, and $u_{y\_mid}$ and $\Delta u_x$ correlate well. These two alternative damage state indicators are therefore well represented by the vertical displacement of the crest midpoint of the embankment. On the other hand, Figure 11b shows that the difference in vertical displacement of the midpoint and edge points ($|\Delta u_y|$) responds differently than the settling of the embankment midpoint.

These results confirm that permanent settling of the midpoint of the crest of an embankment is a relatively good damage state indicator, since it: (i) can represent well some other possible damage state indicators; and (ii) is not sensitive to possible local phenomena at the edges of the embankment. Nevertheless, other representative EDPs should be used if differential settlements are expected to be the driving mechanism behind the damage of an examined embankment.

## 5. Conclusions

The dynamic response of embankments built on soil deposits susceptible to liquefaction was studied through vulnerability analysis. A large set of fragility curves was derived based on the results of numerical calculations using the software package FLAC 2D and the PM4Sand material model for simulating the behavior of the liquefiable layer during dynamic excitation. The width and height of the embankment, thickness and density state of the liquefiable layer, and effect of a 1 m thick crust layer were examined in this parametric study. In order to derive fragility curves, each variation of the model was loaded by at least eight intensities of a set of 30 GMs. Permanent vertical displacement of midpoint at embankment crest was selected as the EDP, while peak ground acceleration at bedrock was used as the IM. Threshold values for assessing three damage states, from minor to extensive deformation of the embankment, were taken from the literature ([14]). According to the presented results, similar tendencies were observed when the criteria for road or railway embankments were applied in the fragility analysis. Generally, all the curves for the railway embankments shifted to the left compared to the curves for the road embankments due to the more severe serviceability limitations of railway transportation.

Despite the limited number of model variations, certain trends in embankment response were identified. Both the height of the embankment and the thickness of the liquefiable sandy layer increase the vulnerability, while increasing the crest width and the relative density of the sandy layer reduced settling at the central top point, thus diminishing the vulnerability of the embankment. A positive effect of the clayey crust layer on top of the liquefiable layer was also observed.

Since it was not clear in advance whether vertical displacement at crest midpoint would be a robust representative parameter for assessing the overall performance of the traffic embankment in the case of seismic loading and liquefiable ground, some correlations between different embankment displacements were analyzed. The results showed that the settling of the crest midpoint correlated well with the maximum settlement of the crest edge points and with the differential horizontal displacements. On the other hand, it was observed that differential vertical settlements could not be reliably predicted based on vertical displacements at the embankment midpoint. Based on that finding, other EDPs (e.g., rotation or differential settlement) should be used when differential vertical displacement is important for assessing the damage state of an embankment.

The results of the presented fragility analyses may be of limited value for specific locations, but they may be relevant for preliminary vulnerability assessments of traffic embankments. The main purpose of this paper was to present the influence of key variables regarding model geometry and material properties on the behavior of embankments on liquefiable ground exposed to seismic excitation by mean of fragility curves. The study is useful for assessing the vulnerability of the entire transport network of a wider area, as time and financial constraints make it difficult to afford detailed analyses of each embankment separately. A generic parametric study is needed to capture the spatial variability of soil properties and geometry of the embankments, the results of which can later be applied to different scenarios in a vulnerability analysis of transport network systems. Fragility curves and other outcomes of the study related to the effects of model variables might serve as an input parameter to an integrated framework of a comprehensive study dealing with infrastructure or traffic network resilience [41,42] as well as overall societal flexibility to recover from a natural disaster. In addition, the outcomes of such studies enable a rapid assessment of vulnerability, which can serve as a starting point for a detailed analysis of critical embankments.

**Author Contributions:** Conceptualization, A.O., M.K., A.V.D.F. and J.L.; methodology, A.O., M.K., A.V.D.F. and J.L.; software, A.O. and A.V.D.F.; validation, A.O. and M.K.; formal analysis, A.O.; investigation, A.O.; resources, A.O. and M.K.; data curation, A.O.; writing—original draft preparation, A.O.; writing—review and editing, A.O., M.K., A.V.D.F. and J.L.; visualization, A.O.; supervision, A.V.D.F. and J.L.; project administration, A.V.D.F. and J.L.; funding acquisition, A.V.D.F. and J.L. All authors have read and agreed to the published version of the manuscript.

**Funding:** The work was supported by the LIQUEFACT project (Assessment and mitigation of liquefaction potential across Europe: A holistic approach to protect structures/infrastructures for improved resilience to earthquake-induced liquefaction disasters), which has received funding from the European Union's Horizon 2020 research and innovation programme under grant agreement No. GAP-700748.

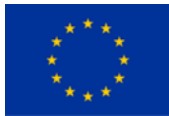

**Conflicts of Interest:** The authors declare no conflict of interest.

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
