# Peer review of "Fragility Assessment of Traffic Embankments Exposed to Earthquake-Induced Liquefaction"

_applsci, doi:10.3390/app10196832_

Round 1

Reviewer 1 Report

In general, this study can be of potential interest to the readers of Applied Sciences.

However, I have two major concerns, and I would appreciate if the authors elaborate more on the following points in their manuscript:

- Originality / filling a research gap: The authors should stress more on what is original and what is routine work in their study. Do they present novel techniques? Do they apply them to a new problem for the first time? Do they obtain novel insights, so far unknown? I am having a hard time recognising much originality in applying well-known soil models in a commercial software to carry out parametric simulations...

- Practical use: The authors should explain better how their results can be used in practise (directly or as a part of a bigger picture). What advantage/usability do their result have? Can their results be used in actual problems/assessments? Can they be used to formulate practical guidelines?

Furthermore, I have a series of important comments, which are expressed below:

At line 136, you state that: "the assumption of independence of the observations may not be strictly true [...]. Nevertheless, Baker [18] suggested that relaxing this assumption would typically make little numerical difference in the estimates of the parameters of the fragility function. Thus, it can be expected that the procedure for the estimation of the parameters of the fragility function will produce effective fragility estimates, even using IDA data". I think the authors should not simply assume this. They should actually demonstrate that relaxing this hypothesis is a viable strategy, and quantify the numerical difference.

In table 1: rather than the min-mean-max visualization, I believe that to capture the actual variability of the values you should present them in terms of mean-standard deviation (and n. of samples). Also, objective and quantitative definitions should be provided for the ds1-2-3 damage level to avoid ambiguities.

About the choice of values in table 2, the authors could provide a justification that the values that they considered actually cover a reasonable range of practical cases. Also, they should justify (with references, even though it is more or less "common" knowledge) why 6, 12, 18 metres in width are actually representative of local, regional and motorways (see line 180).

Line 187. How did you decide that 17 model variations were an adequate number? Why not 30 or 100? Is there a limitation in the computational capacity?

Line 196 (and table 4). How did you decide the precise parameters of the materials? Wouldn't it be better to consider the material properties with a certain degree of uncertainty/inhomogeneity? This would be more representative of reality as, even in artificial slopes/embankments, the material properties cannot be fully controlled.

Line 206. Some words in justification of the material models would be appreciated. Were there any other options and, if so, why were they discarded?

Line 245. Is a set of 30 GMs sufficient? Is this number backed by a guideline? Please elaborate.

Figure 11 and 12. The coefficient of determination (R-squared) is not sufficient alone to express a correlation. The authors should provide p-values for their correlations to demonstrate whether they are significant or insignificant at a defined confidence level.

Line 417. What is your threshold to define a correlation "good"? Do you expect an R-squared of 0.8 to be good and an R-squared of 0.6 to be bad? Please elaborate.

Reviewer 2 Report

Seismic fragility curves for transport infrastructure embankments on liquefiable soil are developed based on numerical modelling. The study is based on previous approaches and engineering assumptions. However, key parameters, such as model geometry and material properties, are investigated. The results are meaningful, the following comments are provided.

Some recent publications on fragility analysis of embankments can be added in the literature, eg.:

McKenna, G., Argyroudis, S. A., Winter, M. G., & Mitoulis, S. A. (2020). Multiple hazard fragility analysis for granular highway embankments: moisture ingress and scour. Transportation Geotechnics, 100431.

Also, it is suggested to elaborate a bit more on how the proposed fragility curves can be useful toward more resilient transport infrastructure, e.g.:

Pitilakis, K., Argyroudis, S., Kakderi, K., & Selva, J. (2016). Systemic vulnerability and risk assessment of transportation systems under natural hazards towards more resilient and robust infrastructures. Transportation Research Procedia14, 1335-1344.

Argyroudis, S. A., Mitoulis, S. A., Hofer, L., Zanini, M. A., Tubaldi, E., & Frangopol, D. M. (2020). Resilience assessment framework for critical infrastructure in a multi-hazard environment: Case study on transport assets. Science of The Total Environment714, 136854.

‘PGA at bedrock was selected as the IMs’. However, PGA at ground surface is commonly used in practice, perhaps the authors can further explain their choice, using examples from the literature.

“other representative EDPs should be used if differential settlements are expected to be the driving mechanism”, please specify which EDPs should be used.

The proposed fragility functions have not been validated on the basis on observations from past events and/or other fragility curves in the literature. Please add such verification or comment.

Round 2

Reviewer 1 Report

The authors have responded to my comments satisfactorily.

I do not have additional comments and believe the manuscript is now acceptable for publication.

Author Response

The authors would like to thank the reviewers (and the editor) once again for all their constructive comments and fast reply.

As reviewer 1 had no further comments, no additional responses were prepared.

The authors would like to thank you again for taking the time to review our paper, and we sincerely hope that your valuable comments/contributions have been taken into account for the final version.

Reviewer 2 Report

The authors have replied to the reviewers' comments, and updated the manuscript. However some comments were not considered in the manuscript. For example the discussion/answers on the number of GM or the material properties can be also briefly included in the manuscript.

In the validation of the fragility curves (lines 348-357) it is not clear if the validated fragility curves correspond to loose or medium dense, and to roads or railways. Also, the observed damage was minor, while the estimated 0.05-0.08 of exceedance probability for minor damage is too low compared to the observed.

Furthermore the probability of being in each damage state should be estimated, see Fig.1 in McKenna et al. 2020 for definitions and Winter et al. 2014 https://link.springer.com/article/10.1007/s10064-014-0570-3 for validation example.
